# Achieving the KS threshold in the general stochastic block model with linearized acyclic belief propagation

**Emmanuel Abbe**
Applied and Computational Mathematics and EE Dept.
Princeton University
eabbe@princeton.edu

**Colin Sandon**
Department of Mathematics
Princeton University
sandon@princeton.edu

## Abstract

The stochastic block model (SBM) has long been studied in machine learning and network science as a canonical model for clustering and community detection. In the recent years, new developments have demonstrated the presence of threshold phenomena for this model, which have set new challenges for algorithms. For the *detection* problem in symmetric SBMs, Decelle et al. conjectured that the so-called Kesten-Stigum (KS) threshold can be achieved efficiently. This was proved for two communities, but remained open for three and more communities. We prove this conjecture here, obtaining a general result that applies to arbitrary SBMs with linear size communities. The developed algorithm is a linearized acyclic belief propagation (ABP) algorithm, which mitigates the effects of cycles while provably achieving the KS threshold in $O(n \ln n)$ time. This extends prior methods by achieving universally the KS threshold while reducing or preserving the computational complexity. ABP is also connected to a power iteration method on a generalized nonbacktracking operator, formalizing the spectral-message passing interplay described in Krzakala et al., and extending results from Bordenave et al.

## 1   Introduction

The stochastic block model (SBM) is widely used as a model for community detection and as a benchmark for clustering algorithms. The model emerged in multiple scientific communities, in machine learning and statistics under the SBM [1, 2, 3, 4], in computer science as the planted partition model [5, 6, 7], and in mathematics as the inhomogeneous random graph model [8]. Although the model was defined as far back as the 80s, mainly studied for the exact recovery problem, it resurged in the recent years due in part to fascinating conjectures on the *detection* problem, established in [9] (and backed in [10]) from deep but non-rigorous statistical physics arguments. For efficient algorithms, the following was conjectured:

**Conjecture 1.** *(See formal definitions below) In the stochastic block model with $n$ vertices, $k$ balanced communities, edge probability $a/n$ inside the communities and $b/n$ across, it is possible to detect communities in polynomial time if and only if*

$$\frac{(a-b)^2}{k(a+(k-1)b)} > 1. \tag{1}$$

In other words, the problem of detecting efficiently communities is conjectured to have a sharp threshold at the above, which is called the Kesten-Stigum (KS) threshold. Establishing such thresholds is of primary importance for the developments of algorithms. A prominent example is Shannon's coding theorem, that gives a sharp threshold for coding algorithms at the channel capacity, and which has led the development of coding algorithms used in communication standards. In the area of

clustering, where establishing rigorous benchmarks is a challenge, the quest of sharp thresholds is likely to also have fruitful outcomes.

Interestingly, classical clustering algorithms do not seem to suffice for achieving the threshold in (1). This includes spectral methods based on the adjacency matrix or Laplacians, as well as SDPs. For standard spectral methods, a first issue is that the fluctuations in the node degrees produce high-degree nodes that disrupt the eigenvectors from concentrating on the clusters. This issue is further enhanced on real networks where degree variations are important. A classical trick is to trim such high-degree nodes [11, 12], throwing away some information, but this does not seem to suffice. SDPs are a natural alternative, but they also stumble before the threshold [13, 14], focusing on the most likely rather than typical clusterings.

Significant progress has already been achieved on Conjecture 1. In particular, the conjecture is set for $k = 2$, with the achievability part proved in [15, 16] and [17], and the impossibility part in [10]. Achievability results were also obtained in [17] for SBMs with multiple communities that satisfy a certain asymmetry condition (see Theorem 5 in [17]). Conjecture 1 remained open for $k \geq 3$.

In their original paper [9], Decelle et al. conjectured that belief propagation (BP) achieves the KS threshold. The main issue when applying BP to the SBM is the classical one: the presence of cycles in the graph makes the behavior of the algorithm difficult to understand, and BP is susceptible to settle down in the wrong fixed points. While empirical studies of BP on loopy graph have shown that convergence still takes place in some cases [18], obtaining rigorous results in the context of loopy graphs remains a long standing challenge for message passing algorithms, and achieving the KS threshold requires precisely running BP to an extent where the graph is not even tree-like. We address this challenge in the present paper, with a linearized version of BP that mitigates cycles.

Note that establishing formally the converse of Conjecture 1 (i.e., that efficient detection is impossible below the threshold) for arbitrary $k$ seems out of reach at the moment, as the problem behaves very differently for small rather than arbitrary $k$. Indeed, except for a few low values of $k$, it is proven in [19, 20] that the threshold in (1) does not coincide with the information-theoretic threshold. Since it is possible to detect below the threshold with non-efficient algorithms, proving formally the converse of Conjecture 1 would require major headways in complexity theory. On the other hand, [9] provides already non-rigourous arguments that the converse hold.

## 1.1 This paper

This paper proves the achievability part of conjecture 1. Our main result applies to a more general context, with a generalized notion of detection that applies to arbitrary SBMs. In particular,

- we show that an approximate belief propagation (ABP) algorithm that mitigates cycles achieves the KS threshold universally. The simplest linearized[1] version of BP is to repeatedly update beliefs about a vertex's community based on its neighbor's suspected communities while avoiding backtrack. However, this only works ideally if the graph is a tree. The correct response to a cycle would be to discount information reaching the vertex along either branch of the cycle to compensate for the redundancy of the two branches. Due to computational issues we simply prevent information from cycling around constant size cycles.
- we show how ABP can be interpreted as a power iteration method on a generalized $r$-nonbacktracking operator, i.e., a spectral algorithm that uses a matrix counting the number of $r$-nonbacktracking walks rather than the adjacency matrix. The random initialization of the beliefs in ABP corresponds to the random vector to which the power iteration is applied, formalizing the connection described in [22]. While using $r = 2$ backtracks may suffice to achieve the threshold, larger backtracks are likely to help mitigating the presence of small loops in networks.

Our results are closest to [16, 17], while diverging in several key parts. A few technical expansions in the paper are similar to those carried in [16], such as the weighted sums over nonbacktracking walks and the SAW decomposition in [16], similar to our compensated nonbacktracking walk counts and Shard decomposition. Our modifications are developed to cope with the general SBM, in particular to compensation for the dominant eigenvalues in the latter setting. Our algorithm complexity is also slightly reduced by a logarithmic factor.

Our algorithm is also closely related to [17], which focuses on extracting the eigenvectors of the standard nonbacktracking operator. Our proof technique is different than the one in [17], so that we can cope with the setting of Conjecture 1. We also implement the eigenvector extractions in a belief propagation fashion. Another difference with [17] is that we rely on nonbacktracking operators of higher orders $r$. While $r = 2$ is arguably the simplest implementation and may suffice for the sole purpose of achieving the KS threshold, a larger $r$ is likely to be beneficial in practice. For example, an adversary may add triangles for which ABP with $r = 2$ would fail while larger $r$ would succeed. Finally, the approach of ABP can be extended beyond the linearized setting to move from detection to an optimal accuracy as discussed in Section 5.

## 2 Results

### 2.1 A general notion of detection

The stochastic block model (SBM) is a random graph model with clusters defined as follows.

**Definition 1.** *For $k \in \mathbb{Z}_+$, a probability distribution $p \in (0,1)^k$, a $k \times k$ symmetric matrix $Q$ with nonnegative entries, and $n \in \mathbb{Z}_+$, we define $\mathrm{SBM}(n, p, Q/n)$ as the probability distribution over ordered pairs $(\sigma, G)$ of an assignment of vertices to one of $k$ communities and an $n$-vertex graph generated by the following procedure. First, each vertex $v \in V(G)$ is independently assigned a community $\sigma_v$ under the probability distribution $p$. Then, for every $v \neq v'$, an edge is drawn in $G$ between $v$ and $v'$ with probability $Q_{\sigma_v, \sigma_{v'}}/n$, independently of other edges. We sometimes say that $G$ is drawn under $\mathrm{SBM}(n, p, Q/n)$ without specifying $\sigma$ and define $\Omega_i = \{v : \sigma_v = i\}$.*

**Definition 2.** *The SBM is called symmetric if $p$ is uniform and if $Q$ takes the same value on the diagonal and the same value off the diagonal.*

Our goal is to find an algorithm that can distinguish between vertices from one community and vertices from another community in a non trivial way.

**Definition 3.** *Let $A$ be an algorithm that takes a graph as input and outputs a partition of its vertices into two sets. $A$ solves detection (or weak recovery) in graphs drawn from $\mathrm{SBM}(n, p, Q/n)$ if there exists $\epsilon > 0$ such that the following holds. When $(\sigma, G)$ is drawn from $\mathrm{SBM}(n, p, Q/n)$ and $A(G)$ divides its vertices into $S$ and $S^c$, with probability $1 - o(1)$, there exist $i, j \in [k]$ such that $|\Omega_i \cap S|/|\Omega_i| - |\Omega_j \cap S|/|\Omega_j| > \epsilon$.*

In other words, an algorithm solves detection if it divides the graph's vertices into two sets such that vertices from different communities have different probabilities of being assigned to one of the sets. An alternate definition (see for example Decelle et al. [9]) requires the algorithm to divide the vertices into $k$ sets such that there exists $\epsilon > 0$ for which there exists an identification of the sets with the communities labelling at least $\max p_i + \epsilon$ of the vertices correctly with high probability. In the 2 community symmetric case, the two definitions are equivalent. In a two community asymmetric case where $p = (.2, .8)$, an algorithm that could find a set containing $2/3$ of the vertices from the large community and $1/3$ of the vertices from the small community would satisfy Definition 3, however, it would not satisfy previous definition due to the biased prior. If all communities have the same size, this distinction is meaningless and we have the following equivalence:

**Lemma 1.** *Let $k > 0$, $Q$ be a $k \times k$ symmetric matrix with nonnegative entries, $p$ be the uniform distribution over $k$ sets, and $A$ be an algorithm that solves detection in graphs drawn from $\mathrm{SBM}(n, p, Q/n)$. Then $A$ also solves detection according to Decelle et al.'s criterion [9], provided that we consider it as returning $k - 2$ empty sets in addition to its actual output.*

*Proof.* Let $(\sigma, G) \sim \mathrm{SBM}(n, p, Q/n)$ and $A(G)$ return $S$ and $S'$. There exists $\epsilon > 0$ such that with high probability (whp) there exist $i, j$ such that $|\Omega_i \cap S|/|\Omega_i| - |\Omega_j \cap S|/|\Omega_j| > \epsilon$. So, if we map $S$ to community $i$ and $S'$ to community $j$, the algorithm classifies at least $|\Omega_i \cap S|/n + |\Omega_j \cap S'|/n = |\Omega_j|/n + |\Omega_i \cap S|/n - |\Omega_j \cap S|/n \geq 1/k + \epsilon/k - o(1)$ of the vertices correctly whp. $\square$

### 2.2 Achieving efficiently and universally the KS threshold

Given parameters $p$ and $Q$ for the SBM, let $P$ be the diagonal matrix such that $P_{i,i} = p_i$ for each $i \in [k]$. Also, let $\lambda_1, ..., \lambda_h$ be the distinct eigenvalues of $PQ$ in order of nonincreasing magnitude.

**Definition 4.** *The signal to noise ratio of* $\mathrm{SBM}(n, p, Q/n)$ *is defined by* $\mathrm{SNR} := \lambda_2^2/\lambda_1$.

**Theorem 1.** *Let* $k \in \mathbb{Z}_+$, $p \in (0,1)^k$ *be a probability distribution, $Q$ be a $k \times k$ symmetric matrix with nonnegative entries, and $G$ be drawn under* $\mathrm{SBM}(n, p, Q/n)$. *If* $\mathrm{SNR} > 1$, *then there exist* $r \in \mathbb{Z}^+$, $c > 0$, *and* $m : \mathbb{Z}^+ \to \mathbb{Z}^+$ *such that* $\mathrm{ABP}(G, m(n), r, c, (\lambda_1, ..., \lambda_h))$ *described in the next section solves detection and runs in* $O(n \log n)$ *time.*

For the symmetric SBM, this settles the achievability part of Conjecture 1, as the condition $\mathrm{SNR} > 1$ reads in this case $\mathrm{SNR} = (\frac{a-b}{k})^2/(\frac{a+(k-1)b}{k}) = (a-b)^2/(k(a+(k-1)b)) > 1$.

## 3 The linearized acyclic belief propagation algorithm (ABP)

### 3.1 Vanilla version

We present first a simplified version of our algorithm that captures the essence of the algorithm while avoiding technicalities required for the proof, described in Section 3.3.

$\mathrm{ABP}^*(G, m, r, \lambda_1)$:

1. For each vertex $v$, randomly draw $x_v$ with a Normal distribution. For all adjacent $v, v'$ in $G$, set $y_{v,v'}^{(1)} = x_{v'}$ and set $y_{v,v'}^{(t)} = 0$ whenever $t < 1$.
2. For each $1 < t \le m$, set

$$z_{v,v'}^{(t-1)} = y_{v,v'}^{(t-1)} - \frac{1}{2|E(G)|} \sum_{(v'',v''') \in E(G)} y_{v'',v'''}^{(t-1)} \tag{2}$$

   for all adjacent $v, v'$. For each adjacent $v, v'$ that are not part of a cycle of length $r$ or less, set

$$y_{v,v'}^{(t)} = \sum_{v'':(v',v'')\in E(G), v'' \ne v} z_{v',v''}^{(t-1)}$$

   and for the other adjacent $v, v'$ in $G$, let the other vertex in the cycle that is adjacent to $v$ be $v'''$, the length of the cycle be $r'$, and set

$$y_{v,v'}^{(t)} = \sum_{v'':(v',v'')\in E(G), v'' \ne v} z_{v',v''}^{(t-1)} - \sum_{v'':(v,v'')\in E(G), v'' \ne v', v'' \ne v'''} z_{v,v''}^{(t-r')}$$

   unless $t = r'$, in which case, set $y_{v,v'}^{(t)} = \sum_{v'':(v',v'')\in E(G), v'' \ne v} z_{v',v''}^{(t-1)} - z_{v''',v}^{(1)}$.
3. Set $y_v' = \sum_{v':(v',v)\in E(G)} y_{v,v'}^{(m)}$ for every $v \in G$ and return $(\{v : y_v' > 0\}, \{v : y_v' \le 0\})$.

**Remarks.** (1) In the $r = 2$ case, one can exit step 2 after the second line. As mentioned above, we rely on a less compact version of the algorithm to prove the theorem, but expect that the above also succeeds at detection as long as $m > 2\ln(n)/\ln(\mathrm{SNR})$.

(2) What the algorithm does if $(v, v')$ is in multiple cycles of length $r$ or less is unspecified as there is no such edge with probability $1 - o(1)$ in the sparse SBM. This can be modified for more general settings, applying the adjustment independently for each such cycle, setting $y_{v,v'}^{(t)} = \sum_{v'':(v',v'')\in E(G), v'' \ne v} z_{v',v''}^{(t-1)} - \sum_{r'=1}^{r} \sum_{v''':(v,v''')\in E(G)} C_{v''',v,v'}^{(r')} \sum_{v'':(v,v'')\in E(G), v'' \ne v', v'' \ne v'''} z_{v,v''}^{(t-r')}$, where $C_{v''',v,v'}^{(r')}$ denotes the number of length $r'$ cycles that contain $v''', v, v'$ as consecutive vertices.

(3) The purpose of setting $z_{v,v'}^{(t-1)}$ as in step (2) is to ensure that the average value of the $y^{(t)}$ is approximately 0, and thus that the eventual division of the vertices into two sets is roughly even. An alternate way of doing this is to simply let $z_{v,v'}^{(t-1)} = y_{v,v'}^{(t-1)}$ and then compensate for any bias of $y^{(t)}$ towards positive or negative values at the end. More specifically, define $Y$ to be the $n \times m$ matrix such that for all $t$ and $v$, $Y_{v,t} = \sum_{v':(v',v)\in E(G)} y_{v,v'}^{(t)}$, and $M$ to be the $m \times m$ matrix such that $M_{i,i} = 1$ and $M_{i,i+1} = -\lambda_1$ for all $i$, and all other entries of $M$ are equal to 0. Then set $y' = YM^{m'}e_m$, where $e_m \in \mathbb{R}^m$ denotes the unit vector with 1 in the $m$-th entry, and $m'$ is a suitable integer.

## 3.2 Spectral implementation

One way of looking at this algorithm for $r = 2$ is the following. Given a vertex $v$ in community $i$, the expected number of vertices $v'$ in community $j$ that are adjacent to $v$ is approximately $e_j \cdot PQe_i$. For any such $v'$ the expected number of vertices in community $j'$ that are adjacent to $v'$ not counting $v$ is approximately $e_{j'} \cdot PQe_j$, and so on. In order to explore this connection, define the graph's nonbacktracking walk matrix $W$ as the $2|E(G)| \times 2|E(G)|$ matrix such that for all $v \in V(G)$ and all distinct $v'$ and $v''$ adjacent to $v$, $W_{(v,v''),(v',v)} = 1$, and all other entries in $W$ are 0.

Now, let $w$ be an eigenvector of $PQ$ with eigenvalue $\lambda_i$ and $\overline{w} \in \mathbb{R}^{2|E(G)|}$ be the vector such that $\overline{w}_{(v,v')} = w_{\sigma_{v'}}/p_{\sigma_{v'}}$ for all $(v, v') \in E(G)$. For any small $t$, we would expect that $\overline{w} \cdot W^t \overline{w} \approx \lambda_i^t \|\overline{w}\|_2^2$, which strongly suggests that $\overline{w}$ is correlated with an eigenvector of $W$ with eigenvalue $\lambda_i$. For any such $w$ with $i > 1$, dividing $G$'s vertices into those with positive entries in $\overline{w}$ and those with negative entries in $\overline{w}$ would put all vertices from some communities in the first set, and all vertices from the other communities in the second. So, we suspect that an eigenvector of $W$ with its eigenvalue of second greatest magnitude would have entries that are correlated with the corresponding vertices' communities.

We could simply extract this eigenvector, but a faster approach would be to take a random vector $y$ and then compute $W^m y$ for some suitably large $m$. That will be approximately equal to a linear combination of $W$'s dominant eigenvectors. Its dominant eigenvector is expected to have an eigenvalue of approximately $\lambda_1$ and to have all of its entries approximately equal, so if instead we compute $(W - \frac{\lambda_1}{2|E(G)|}J)^m y$ where $J$ is the vector with all entries equal to 1, the component of $y$ proportional to $W$'s dominant eigenvector will be reduced to negligable magnitude, leaving a vector that is approximately proportional to $W$'s eigenvector of second largest eigenvalue. This is essentially what the ABP algorithm does for $r = 2$.

This vanilla approach does however not extend obviously to the case with multiple eigenvalues. In such cases, we will have to subtract multiples of the identity matrix instead of $J$ because we will not know enough about $W$'s eigenvectors to find a matrix that cancels out one of them in particular. These are significant challenges to overcome to prove the general result and Conjecture 1.

For higher values of $r$, the spectral view of ABP can be understood as described above but introducing the following generalized nonbacktracking operator as a replacement to $W$:

**Definition 5.** *Given a graph, define the $r$-nonbacktracking matrix $W^{(r)}$ of dimension equal to the number of $r-1$ directed paths in the graph and with entry $W^{(r)}_{(v_1,v_2,...,v_r),(v'_1,v'_2,...,v'_r)}$ equal to 1 if $v'_{i+1} = v_i$ for each $1 \leq i < r$ and $v'_1 \neq v_r$, and equal to 0 otherwise.*

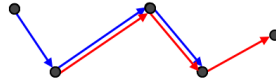

Figure 1: Two paths of length 3 that contribute to an entry of 1 in $W^{(4)}$.

## 3.3 Full version

The main modifications in the proof are as follows. First, at the end we assign vertices to sets with probabilities that scale linearly with their entries in $y'$ instead of simply assigning them based on the signs of their entries. This allows us to convert the fact that the average values of $y'_v$ for $v$ in different communities is different into a detection result. Second, we remove a small fraction of the edges from the graph at random at the beginning of the algorithm (the graph-splitting step), defining $y''_v$ to be the sum of $y'_{v'}$ over all $v'$ connected to $v$ by paths of a suitable length with removed edges at their ends in order to eliminate some dependency issues. Also, instead of just compensating for $PQ$'s dominant eigenvalue, we also compensate for some of its smaller eigenvalues, and subtract multiples of $y^{(t-1)}$ from $y^{(t)}$ for some $t$ instead of subtracting the average value of $y^{(t)}$ from all of its entries for all $t$. We refer to [19] for the full description of the algorithm. Note that while it is easier to prove that the ABP algorithm works, the ABP* algorithm should work at least as well in practice.

## 4 Proof technique

For simplicity, consider first the two community symmetric case. Consider determining the community of $v$ using belief propagation, assuming some preliminary guesses about the vertices $t$ edges away from it, and assuming that the subgraph of $G$ induced by the vertices within $t$ edges of $v$ is a tree. For any vertex $v'$ such that $d(v, v') < t$, let $C_{v'}$ be the set of the children of $v'$. If we believe based on either our prior knowledge or propagation of beliefs up to these vertices that $v''$ is in community 1 with probability $\frac{1}{2} + \frac{1}{2}\epsilon_{v''}$ for each $v'' \in C_{v'}$, then the algorithm will conclude that $v'$ is in community 1 with a probability of

$$\frac{\prod_{v'' \in C_{v'}}(\frac{a+b}{2} + \frac{a-b}{2}\epsilon_{v''})}{\prod_{v'' \in C_{v'}}(\frac{a+b}{2} + \frac{a-b}{2}\epsilon_{v''}) + \prod_{v'' \in C_{v'}}(\frac{a+b}{2} - \frac{a-b}{2}\epsilon_{v''})}.$$

If all of the $\epsilon_{v''}$ are close to 0, then this is approximately equal to (see also [9, 22])

$$\frac{1 + \sum_{v'' \in C_{v'}} \frac{a-b}{a+b}\epsilon_{v''}}{2 + \sum_{v'' \in C_{v'}} \frac{a-b}{a+b}\epsilon_{v''} + \sum_{v'' \in C_{v'}}(-1)\frac{a-b}{a+b}\epsilon_{v''}} = \frac{1}{2} + \frac{a-b}{a+b}\sum_{v'' \in C_{v'}} \frac{1}{2}\epsilon_{v''}.$$

That means that the belief propagation algorithm will ultimately assign an average probability of approximately $\frac{1}{2} + \frac{1}{2}(\frac{a-b}{a+b})^t \sum_{v'':d(v,v'')=t} \epsilon_{v''}$ to the possibility that $v$ is in community 1. If there exists $\epsilon$ such that $E_{v'' \in \Omega_1}[\epsilon_{v''}] = \epsilon$ and $E_{v'' \in \Omega_2}[\epsilon_{v''}] = -\epsilon$ (recall that $\Omega_i = \{v : \sigma_v = i\}$), then on average we would expect to assign a probability of approximately $\frac{1}{2} + \frac{1}{2}\left(\frac{(a-b)^2}{2(a+b)}\right)^t \epsilon$ to $v$ being in its actual community, which is enhanced as $t$ increases when $\mathrm{SNR} > 1$. Note that since the variance in the probability assigned to the possibility that $v$ is in its actual community will also grow as $\left(\frac{(a-b)^2}{2(a+b)}\right)^t$, the chance that this will assign a probability of greater than $1/2$ to $v$ being in its actual community will be $\frac{1}{2} + \Theta\left(\left(\frac{(a-b)^2}{2(a+b)}\right)^{t/2}\right)$.

One idea for the initial estimate is to simply guess the vertices' communities at random, in the expectation that the fractions of the vertices from the two communities assigned to a community will differ by $\theta(1/\sqrt{n})$ by the Central Limit Theorem. Unfortunately, for any $t$ large enough that $\left(\frac{(a-b)^2}{2(a+b)}\right)^{t/2} > \sqrt{n}$, we have that $\left(\frac{(a+b)}{2}\right)^t > n$ which means that our approximation breaks down before $t$ gets large enough to detect communities. In fact, $t$ would have to be so large that not only would neighborhoods not be tree like, but vertices would have to be exhausted.

One way to handle this would be to stop counting vertices that are $t$ edges away from $v$, and instead count each vertex a number of times equal to the number of length $t$ paths from $v$ to it.[2] Unfortunately, finding all length $t$ paths starting at $v$ can be done efficiently enough only for values of $t$ that are smaller than what is needed to amplify a random guess to the extent needed here. We could instead calculate the number of length $t$ walks from $v$ to each vertex more quickly, but this count would probably be dominated by walks that go to a high degree vertex and then leave and return to it repeatedly, which would throw the calculations off. On the other hand, most reasonably short nonbacktracking walks are likely to be paths, so counting each vertex a number of times equal to the number of nonbacktracking walks of length $t$ from $v$ to it seems like a reasonable modification. That said, it is still possible that there is a vertex that is in cycles such that most nonbacktracking walks simply leave and return to it many times. In order to mitigate this, we use $r$-nonbacktracking walks, walks in which no vertex reoccurs within $r$ steps of a previous occurrence, such that walks cannot return to any vertex more than $t/r$ times.

Unfortunately, this algorithm would not work because the original guesses will inevitably be biased towards one community or the other. So, most of the vertices will have more $r$-nonbacktracking walks of length $t$ from them to vertices that were suspected of being in that community than the other. One way to deal with this bias would be to subtract the average number of $r$-nonbacktracking walks to vertices in each set from each vertex's counts. Unfortunately, that will tend to undercompensate for the bias when applied to high degree vertices and overcompensate for it when applied to low

degree vertices. So, we modify the algorithm that counts the difference between the number of $r$-nonbacktracking walks leading to vertices in the two sets to subtract off the average at every step in order to prevent a major bias from building up.

One of the features of our approach is that it extends fairly naturally to the general SBM. Despite the potential presence of more than 2 communities, we still only assign one value to each vertex, and output a partition of the graph's vertices into two sets in the expectation that different communities will have different fractions of their vertices in the second set. One complication is that the method of preventing the results from being biased towards one comunity does not work as well in the general case. The problem is, by only assigning one value to each vertex, we compress our beliefs onto one dimension. That means that the algorithm cannot detect biases orthogonal to that dimension, and thus cannot subtract them off. So, we cancel out the bias by subtracting multiples of the counts of the numbers of $r$-nonbacktracking walks of some shorter length that will also have been affected by it.

More concretely, we assign each vertex an initial value, $x_v$, at random. Then, we compute a matrix $Y$ such that for each $v \in G$ and $0 \le t \le m$, $Y_{v,t}$ is the sum over all $r$-nonbacktracking walks of length $t$ ending at $v$ of the initial values associated with their starting vertices. Next, for each $v$ we compute a weighted sum of $Y_{v,1}, Y_{v,2}, ..., Y_{v,m}$ where the weighting is such that any biases in the entries of $Y$ resulting from the initial values should mostly cancel out. We then use these to classify the vertices.

*Proof outline for Theorem 1.* If we were going to prove that ABP* worked, we would proba- bly define $W_{r[S]}((v_0, ..., v_m))$ to be 1 if for every consecutive subsequence $(i_1, ..., i_{m'}) \subseteq S$, we have that $v_{i_1 - 1}, ..., v_{i_{m'}}$ is a $r$-nonbacktracking walk, and 0 otherwise. Next, de- fine $W_r((v_0, ..., v_m)) = \sum_{S \subseteq (1,...,m)} (-2|E(G)|)^{-|S|} W_{r[S]}((v_0, ..., v_m))$ and $W_m(x, v) = \sum_{v_0, ..., v_m \in G: v_m = v} x_{v_0} W_r((v_0, ..., v_m))$, and we would have that $y'_v = W_m(x, v)$ for $x$ and $y'$ as in $ABP^*$. As explained above, we rely on a different approach to cope with the general SBM.

In order to prove that the algorithm works, we make the following definitions.

**Definition 6.** *For any $r \ge 1$ and series of vertices $v_0, ..., v_m$, let $W_r((v_0, ..., v_m))$ be 1 if $v_0, ..., v_m$ is an $r$-nonbacktracking walk and 0 otherwise. Also, for any $r \ge 1$, series of vertices $v_0, ..., v_m$ and $c_0, ..., c_m \in \mathbb{R}^{m+1}$, let*

$$W_{(c_0,...,c_m)[r]}((v_0, ..., v_m)) = \sum_{(i_0,...,i_{m'}) \in (0,...,m)} \left( \prod_{i \notin (i_0,...,i_{m'})} (-c_i/n) \right) W_r((v_{i_0}, v_{i_1}, ..., v_{i_{m'}})).$$

*In other words, $W_{(c_0,...,c_m)[r]}((v_0, ..., v_m))$ is the sum over all subsequences of $(v_0, ..., v_m)$ that form $r$-nonbacktracking walks of the products of the negatives of the $c_i/n$ corresponding to the elements of $(v_0, ..., v_m)$ that are not in the walks. Finally, let*

$$W_{m/\{c_i\}}(x, v) = \sum_{v_0,...,v_m \in G: v_m = v} x_{v_0} W_{(c_0,...,c_m)[r]}((v_0, ..., v_m)).$$

The reason these definitions are important is that for each $v$ and $t$, we have that

$$Y_{v,t} = \sum_{v_0,...,v_t \in G: v_t = v} x_{v_0} W_r((v_0, ..., v_t))$$

and $y_v^{(m)}$ is equal to $W_{m/\{c_i\}}(x, v)$ for suitable $(c_0, ..., c_m)$. For the full ABP algorithm, both terms in the above equality refer to $G$ as it is after some of its edges are removed at random in the 'graph splitting' step (which explains the presence of $1 - \gamma$ factors in [19]). One can easily prove that if $v_0, ..., v_t$ are distinct, $\sigma_{v_0} = i$ and $\sigma_{v_t} = j$, then

$$E[W_{(c_0,...,c_t)[r]}((v_0, ..., v_t))] = e_i \cdot P^{-1}(PQ)^t e_j / n^t,$$

and most of the rest of the proof centers around showing that $W_{(c_0,...,c_m)[r]}((v_0, ..., v_m))$ such that $v_0, ..., v_m$ are not all distinct do not contribute enough to the sums to matter. That starts with a bound on $|E[W_{(c_0,...,c_m)[r]}((v_0, ..., v_m))]|$ whenever there is no $i, j \ne j'$ such that $v_j = v_{j'}$, $|i - j| \le r$, and $c_i \ne 0$; and continues with an explanation of how to re-express any $W_{(c_0,...,c_m)[r]}((v_0, ..., v_m))$ as a linear combination of expressions of the form $W_{(c'_0,...,c'_{m'})[r]}((v'_0, ..., v'_{m'}))$ which have this property.

Then we use these to prove that for suitable $(c_0, ..., c_m)$, the sum of $|E[W_{(c_0,...,c_m)[r]}((v_0, ..., v_m))]|$ for all sufficiently repetitive $(v_0, ..., v_m)$ is sufficiently small. Next, we observe that

$$W_{(c_0,...,c_m)[r]}((v_0, ..., v_m))W_{(c_0'',...,c_m'')[r]}((v_0'', ..., v_m''))$$
$$= W_{(c_0,...,c_m,0,...,0,c_m'',...,c_0'')[r]}((v_0, ..., v_m, u_1, ..., u_r, v_n'', ...v_0''))$$

if $u_1, ..., u_r$ are new vertices that are connected to all other vertices, and use that fact to translate bounds on expected values to bounds on variances.

That allows us to show that if $m$ and $(c_0, ..., c_m)$ have the appropriate properties and $w$ is an eigenvector of $PQ$ with eigenvalue $\lambda_j$ and magnitude 1, then with high probability

$$|\sum_{v \in V(G)} w_{\sigma_v}/p_{\sigma_v} W_{m/\{c_i\}}(x, v)| = O(\sqrt{n} \prod_{0 \leq i \leq m} |\lambda_j - c_i| + \frac{\sqrt{n}}{\log(n)} \prod_{0 \leq i \leq m} |\lambda_s - c_i|)$$

and

$$|\sum_{v \in V(G)} w_{\sigma_v}/p_{\sigma_v} W_{m/\{c_i\}}(x, v)| = \Omega(\sqrt{n} \prod_{0 \leq i \leq m} |\lambda_j - c_i|).$$

We also show that under appropriate conditions $\mathrm{Var}[W_{m/\{c_i\}}(x, v)] = O((1/n) \prod_{0 \leq i \leq m} (\lambda_s - c_i)^2)$.

Together, these facts would allow us to prove that the differences between the average values of $W_{m/\{c_i\}}(x, v)$ in different communities are large enough relative to the variance of $W_{m/\{c_i\}}(x, v)$ to let us detect communities, except for one complication. Namely, these bounds are not quite good enough to rule out the possibility that there is a constant probability scenario in which the empirical variance of $\{W_{m/\{c_i\}}(x, v)\}$ is large enough to disrupt our efforts at using $W_{m/\{c_i\}}(x, v)$ for detection. Although we do not expect this to actually happen, we rely on the graph splitting step described in Section 3.3 to discard this potential scenario. $\square$

## 5 Conclusions and extensions

This algorithm is intended to classify vertices with an accuracy nontrivially better than that attained by guessing randomly, but it is not hard to convert this to an algorithm that classifies vertices with optimal accuracy. Once one has reasonable initial guesses of which communities the vertices are in, one can simply run full belief propagation on these guesses. This requires bridging the gap from dividing the vertices into two sets that are correlated with their communities in an unknown way, and assigning each vertex a nontrivial probability distribution for how likely it is to be in each community.

One way to do this is to divide $G$'s vertices into those that have positive and negative values of $y'$, and divide its directed edges into those that have positive and negative values of $y^{(m)}$. We would generally expect that edges from vertices in different communities will have different probabilities of corresponding to positive values of $y^{(m)}$. Now, let $d'$ be the largest integer such that at least $\sqrt{n}$ of the vertices have degree at least $d'$, let $S$ be the set of vertices with degree exactly $d'$, and for each $v \in S$, let $\xi_v = |\{v' : (v, v') \in E(G), y'_{(v,v')} > 0\}|$. We would expect that for any given community $i$, the probability distribution of $\xi_v$ for $v \in \Omega_i$ would be essentially a binomial distribution with parameters $d'$ and some unknown probability. So, compute probabilities such that the observed distribution of values of $\xi_v$ approximately matches the appropriate weighted sum of $k$ binomial distributions.

Next, go through all identifications of the communities with these binomial distributions that are consistent with the community sizes and determine which one most accurately predicts the connectivity rates between vertices that have each possible value of $\xi$ when the edge in question is ignored, and treat this as the mapping of communities to binomial distributions. Then, for each adjacent $v$ and $v'$, determine the probability distribution of what community $v$ is in based on the signs of $y'_{(v'',v)}$ for all $v'' \neq v'$. Finally, use these as the starting probabilities for BP with a depth of $\ln(n)/3\ln(\lambda_1)$.

**Acknowledgments**

This research was supported by NSF CAREER Award CCF-1552131 and ARO grant W911NF-16-1-0051.

## Footnotes

[1]Other forms of approximate message passing algorithms have been studied for dense graphs, in particular [21] for compressed sensing.

[2]This type of approach is considered in [23].

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
