[Reviews · NeurIPS 2016]

Reviewer 1

Summary

This work is about the community detection threshold in stochastic block models (SBM). It deals with arbitrary SBM with $k$ communities (i.e. the communities need not to have equal proportion, the interactions between groups is not restricted to within-groups and inter-groups values and $k$ may be larger than 2). The paper establishes that a linearized acyclic belief propagation algorithm achieves the Kesten-Stigum (KS) threshold for group detection. Here, detection is defined in a weaker sense than the previous definition from Decelle et al. In the specific case of $k=2$ and equal group proportions it boils down to Decelle et al.'s definition but is weaker in general.

Qualitative Assessment

Major comments: - As mentioned by the authors in the introduction, the SBM is widely used throughout many areas, and not only for detecting communities. The authors introduce no condition on the parameter matrix Q so that their setup includes (in principle) cases as diverse as communities, anti-communities (graphs of a bipartite type) and any other type of connectivity structure. However, I believe that their algorithm (based on belief propagation and thus on the community structure) will reveal only communities. If I am right, they should specify where this underlying assumption comes into play. Is it only that definition 3 becomes useless when the structure is not the one of communities? - Most of the arguments given here are not understandable to someone not familiar to the field. For e.g. on line 186-187: ‘For any small t, we would expect that \bar w. W^t \bar w is almost equal to \lambda_i^t \|\bar w \|_2^2’. Why is this the case? Maybe you could refer to some textbooks to ease the understanding. Minor comments: - The bibliography needs to be corrected. I believe Ref [12] appeared in Probab. Theory Relat. Fields (2015, vol 162:431–461) with a slightly different title. Authors ordering is wrong for ref [5]. Ref [15] also appeared in PTRF in 2015. Ref [16] appeared in STOC'16. Ref [19] has appeared in FOCS 2015, etc. - please avoid usage of undefined acronyms (e.g. SDP, SAW,…) - End of section 2.1 uses notation introduced at the beginning of section 2.2 (matrix P and eigenvalue \lambda_1). It would be more convenient to move this paragraph. - Theorem 1: notation ABP(G,m(n),r,c,\lambda_1,\dots,\lambda_h)) only appears in the next section. - Remark 1: ‘one should use variable that are accurate to within a factor of…’: to which variable are you referring here? Also the term ln(\lambda_1^2/\lambda_2^2) could be simplified into 2ln(\lambda_1/\lambda_2) - Algorithm description: as the graph G is not directed, the algorithm description should be cautious when referring to edge (v,v’) \in E(G) as here these are directed edges. This remark is valid for the vanilla version as well as the full version (and also the supplementary material). - line 296: ‘whenever there is no repeated vertex within r of a nonzero c_i’. What do you mean? Typos: - line 81-82: ’SBM model’ is redundant. - line 286: do you mean a ‘r-nonbacktracking walk’? - line 288: the sum is over v_0, \dots v_t but v_t has no meaning here. Should be v_m? - line 304: second factor on the right-hand side: is this \lambda_s or \lambda_j? Same applies on line 306. In Supplementary material: - Before Theorem 1: ’SBP algorithm’ should be ABP. - Corollary 1: Notation ‘SBM(n,k,a,b)’ -while easily understandable- has never been defined.

Confidence in this Review

1-Less confident (might not have understood significant parts)


Reviewer 2

Summary

The author prove the Decelle conjecture that the communities can be detected in the stochastic block model (a classical model for structured graphs) starting from the Kesten-Stigum threshold, which was believed indeed to be the algorithmic limit beyond which any polynomial time algorithm would fail. This is achieved by introducing an analyzing a linear "spectral-like" algorithm.

Qualitative Assessment

The author(s ?) prove an important conjecture in the field of community detection and stochastic block modeling, introducing effective new tools along the way, and generalizing the results of Krzakala et al, on the one hand, and of Bordenave et al, on the other. The paper is very technical but the authors try hard to explain their proof technics. I particularly like reading Section 4. Note that, until line 245 the sketch of the proof is indeed exactly similar to the heuristic explanation for the success of BP in Decelle et al (maybe this should be noted ?). The difficulty is further explained in 246-251, and then solutions are proposed (and turned out to be wrong) until the actual approach is explained starting from 275. The difficulty are well explained. Perhaps a more graphical explanation of the construction would be helpful to understand sec 3.2. I do not find Figure 1 to be particularly eloquent. In fact, maybe a graphical illustration of both the construction of the operator would be useful. Quick note "This formalizes the connection described in [22]. While using r = 2 backtracks may suffice to achieve the threshold, larger backtracks are likely to help mitigating the presence of small loops in networks." -> This might be true for the proof, however the methods in [22] seems to work until the KS threshold. Can the authors clarify?

Confidence in this Review

2-Confident (read it all; understood it all reasonably well)


Reviewer 3

Summary

This paper proves a conjecture for the community detection problem called SBM. For SBM, in 2013, the necessary and sufficient condition are shown for binary symmetric cases. However, it was an open question when the number of communities are bigger than 3. There was only a conjecture for the question. This paper proposes a belief propagation algorithm called ABP and shows the conjecture using ABP.

Qualitative Assessment

Pros. -This paper has theoretical values. They show an open problem and the proof is very beautiful! the proof looks correct (I cannot check all the details). -It is well explained why the other algorithms are not possible to show the conjecture in Section 4. Cons. -Although this paper has values for theory, the practical value is very limited. -They do not show that the proposed algorithm is optimal in terms of the number of misclassified nodes, but show only for that the proposed algorithm can be (little) better than random guessing whenever it is possible. It would be nice if the authors can explain why this is an important in practice. -The algorithm is not universal. The algorithm find a partition which separate nodes into two communities for all k. I agree this is enough to show the detectability. However, when the parameters a and b satisfy (a-b)^2/k(a+(k-1)b)>>1, this is very bad. Overall, I love their theoretical results. However, the practical value is too narrow.

Confidence in this Review

3-Expert (read the paper in detail, know the area, quite certain of my opinion)


Reviewer 4

Summary

The authors proposed an approximate belief propagation algorithm that achieves the detectable threshold of the vertex communities in stochastic block models. The efficient algorithm can also be formulated as a power iteration on a generalized r-nonbacktracking matrix, connecting to the spectral interpretation of the problem.

Qualitative Assessment

While the paper deals with a important problem with technical quality, the presentation of the main paper leaves a lot to be desired. The structure is disruptive with major obstacles to understanding. In fact, the appendix has a much better flow in terms of presenting motivations of theorems, introducing necessary terms and lemmas. The main paper looks like a rushed truncation of the full version. In section 2.2, the main Theorem 1 is stated without first introducing the the Acronyms ABP algorithm. Theorem 2 looks more like a Corollary of Theorem 1. The simplified vanilla version of the belief propagation ABP* actually make it more difficult to understand with subtle differences. If ABP* only exists to shorten the paper, it should be removed. Figure 1 is never referred to in the main text. Toy examples like these could in fact be very helpful in explaining the intuitions behind the derivations. For example, the long and winded explanation on page 7 could be proved with illustrations, help explaining the motivation and intuition behind the algorithm. Definition 6 is also quite confusing. W with different subscript combinations needs to be better explained. Acronyms SDP on line 36 was never properly introduced. The paper ends abruptly without proper conclusion and discussions.

Confidence in this Review

2-Confident (read it all; understood it all reasonably well)


Reviewer 5

Summary

This is an important paper. The block model is a popular technique for clustering nodes of a network into communities, with interest across a broad range of disciplines. Based on physics techniques, Decelle et al. conjectured that belief propagation succeeds all the way down to the so-called Kesten-Stigum threshold. This was proved for k=2 groups (using a modified version of BP) by Mossel, Neeman, and Sly, but the case of arbitrary k was open until now. This paper shows that a linear variant of BP, initialized with random Gaussians, succeeds all the way to the KS threshold; this variant can also be viewed as powering the non-backtracking matrix of ref. [22] (with some additional modifications to handle short cycles). It’s important to point out that going from k=2 to larger k is not incremental; indeed, multiple new phenomena appear for k > 3, including a conjectured “hard but detectable” regime below the KS threshold where detection is information-theoretically possible, but is conjectured to be computationally hard. One question is whether the algorithm and analysis could be simplified by first destroying all short cycles; this would only require removing O(1) edges w.h.p., so it shouldn’t do too much damage to the graph (unlike removing high-degree vertices)

Qualitative Assessment

It’s a little awkward to state Conjecture 1 without first defining a, b, or k. It would also be more readable if you pointed out that the denominator is k^2 times the average degree c. Many previous works write this inequality as |a-b| > k \sqrt{c} (although I understand you want to phrase it as a signal-to-noise ratio). “non-formal evidence” -> “non-rigorous arguments” reduced _by_ a logarithmic factor “Since it is possible to detect below the threshold with non-efficient algorithms”: you should add a reference here to Banks et al., Proc. COLT 2016, which determines the information-theoretic threshold up to a constant (at least in the typical regime where a/b = O(1) as k increases) “This avoids building the nonbacktracking matrix whose dimension grows with the number of edges”: ref. [22] points out that there is a 2n-dimensional matrix which shares the nontrivial parts of the spectrum of the non-backtracking operator (indeed, that’s the matrix they use for their clustering algorithm). So I’m not sure that this is really a problem. p.5: one should use variables (plural) fix “approxiamtely” p.6: the linearization of BP around the trivial fixed point here looks a lot like calculations done previously in refs. [22] and [11]. I would add a reference to make this clear.

Confidence in this Review

3-Expert (read the paper in detail, know the area, quite certain of my opinion)


Reviewer 6

Summary

The paper proves that for over 1 values of the signal to noise ratio, there is an algorithm that runs in polynomial time and "correctly" detects the communities in the general case of the stochastic block model. A corollary of the results for the symmetric stochastic block models shows the "easy" direction of the conjecture related to the o-called Kesten-Stigum threshold for an arbitrary number of communities in the stochastic block model. The algorithm is a generalization of the message passing algorithm, which the authors call the "approximate belief propagation".

Qualitative Assessment

The paper is well-written and to the best of my knowledge contains relatively novel results, and therefore suitable for publication. My one concern is that in my opinion the technicality of the paper is above the expected level for NIPS. This leads to hand-waiving in proofs in the paper, and a very long supplementary material (which I admit I did not read). Thus I am not sure if a conference paper of this form is a suitable venue for publishing this paper. My other concern is that the papers that are mostly related to this one are generally in preprint form (and not published). I did not find anywhere where a result has been used directly from the those papers, but it makes the reader uneasy about the correctness of algorithms and results. I was slightly confused about the relationship between the definition of "solving the detection problem" in this paper and other definitions existing in the literature. I see that the definition in this paper implies the Decelles's criterion, which suffices for the provided theorem as the theorem proves only one direction of the main conjecture, but what about the other direction and the overall relationship? I also think the paper should make clear which r and m will be chosen in the main results of the paper. The paper also seems to implicitly assume that information will not go around a cycle. Have you, for example, tried to show the results for chordless graphs?

Confidence in this Review

2-Confident (read it all; understood it all reasonably well)